# Role of Homeobox A1 in Airway Epithelial Generation from Human Airway Basal Cells

**DOI:** 10.3390/cells14070549

**Published:** 2025-04-05

**Authors:** Mohsen Tabasi, Nathaniel Chen, Umadevi Sajjan

**Affiliations:** 1Center for Inflammation and Lung Research, Lewis-Katz Medical School, Temple University, Philadelphia, PA 19140, USA; mohsen.tabasi@temple.edu (M.T.); nathaniel.chen0001@temple.edu (N.C.); 2Department of Microbiology, Immunology and Inflammation, Lewis-Katz Medical School, Temple University, Philadelphia, PA 19140, USA; 3Department of Thoracic Medicine and Surgery, Temple University Health System, Philadelphia, PA 19140, USA

**Keywords:** airway epithelial repair, goblet cell metaplasia, cell polarization, occludin, cell proliferation, mucociliary-differentiated cell cultures

## Abstract

Airway basal cells from chronic obstructive pulmonary disease patients show a reduction in *HOXA1* expression and generate an abnormal airway epithelium. Because the specific role of HOXA1 in airway basal cells is not known, we investigated the contribution of HOXA1 in the generation of the airway epithelium, which depends on basal cell proliferation, polarization, and differentiation. Airway stem cells were transduced with an inducible *HOXA1* shRNA lentivector to knock down *HOXA1* in either proliferating cells or100% confluent cells. The bronchial epithelium expresses HOXA1 near the basement membrane, likely representing basal cells. *HOXA1* knockdown in proliferating basal cells attenuated cell proliferation. *HOXA1* knockdown in confluent monolayers of basal cells generated an abnormal airway epithelium characterized by goblet cell hyperplasia and an inflammatory phenotype. Compared to the control, HOXA1 knockdown cells showed a decrease in transepithelial resistance, localization of occludin and E-cadherin to the intercellular junctions, reduced expression of occludin but not E-cadherin, and increased expression of TNF-α. Blocking TNF-α increased the expression of occludin in HOXA1 K/D cells. Based on these results, we conclude that HOXA1 plays an important role in cell proliferation, polarization, and differentiation, which are essential steps in airway epithelial generation. Additionally, HOXA1 may regulate occludin expression by inhibiting TNF-α expression.

## 1. Introduction

The airway epithelium lining the conductive airway serves as a crucial first line of defense, protecting the lungs from environmental insults and orchestrating innate and adaptive immune responses. The epithelium in the conductive airways is pseudostratified and made up of five main cell types, ciliated, goblet, club, basal, and suprabasal cells, along with other rare cell types [1]. Goblet cells secrete mucins, which form a protective layer on the surface of the airway epithelium and captures the inhaled particulates, including pathogens. The unidirectional beating of the cilia clears the captured particulates along with the secreted mucus. Thus, ciliated and goblet cells play a major role in the mucociliary clearance function of the airway epithelium to remove inhaled particulates. The airway epithelial cells respond to viral and bacterial infections by expressing antimicrobial peptides and antiviral type I and type III interferons to clear the infecting pathogens [2]. These cells also express pro-inflammatory cytokines in response to pathogens and allergens to orchestrate the lungs’ innate and adaptive immunity to maintain homeostasis. Patients with chronic obstructive pulmonary disease show goblet cell metaplasia in the airways, with an increase in goblet cells at the expense of ciliated cells. Increased mucus secretion by the goblet cells and a decrease in the number of ciliated cells lead to an attenuated mucociliary clearance function, leading to airway obstruction [3].

Basal cells are tissue-specific stem cells [4,5], and the suprabasal cells derived from basal cells are committed to differentiating into ciliated, goblet, club, or other rare cell types. During homeostasis, the basal cells remain relatively quiescent, but they migrate and rapidly proliferate in response to injury to close the wound and regenerate the mucociliary-differentiated airway epithelium with all of the major and rare cell types [4,6,7]. Previously, we and others have demonstrated that basal cells isolated from the airways of patients with chronic obstructive pulmonary disease (COPD) show defects in polarization of the cells, leading to the generation of a structurally and functionally abnormal airway epithelium [8,9,10,11]. Compared to an airway epithelium generated by normal basal cells, the airway epithelium generated from COPD basal cells showed less ciliated cells and more goblet and basal cells [10,11]. Functionally, mucociliary-differentiated COPD airway epithelial cell cultures show exaggerated inflammatory and antiviral interferon responses to rhinovirus [10,12] and attenuated antimicrobial responses to bacterial infections [13,14]. These observations clearly indicate that COPD airway basal cells have defects in generating a structurally and functionally normal airway epithelium, but the mechanisms are not well-understood. Recently, by comparing the transcriptomic profiles of normal and COPD basal cells, we identified the downregulation of several transcription factors involved in tissue development and differentiation, such as *homeobox (HOX) A1*, *HOXB2*, *E64-like ETS transcription factor (ELF)3*, and *ELF5* [15,16]. Notably, while *HOXB2* expression increases during the polarization of cells, *HOXA1* expression remains relatively at the same level during the generation of the airway epithelium, indicating its role in basal cell proliferation and differentiation [15]. Intriguingly, *HOXA1* gene expression was found to be downregulated by two to three logs in COPD basal cells compared to normal cells. Additionally, previous research has demonstrated the expression of *HOXA1* in human adult lungs [17]. However, the specific role of *HOXA1* in the generation of the airway epithelium remains unclear to date.

*HOXA1* belong to a conserved family of homeodomain-containing transcription factors. In mammals, there are 39 *HOX* genes arranged collinearly, which can be categorized into 13 paralogues based on sequence similarity and positioning within the cluster. *HOX* genes are essential for proper anteroposterior axial patterning during embryonic development [18]. *HOXA1* is one of the earliest *HOX* genes expressed during embryonic development, and it is involved in the development of the inner ear and the heart and the segmentation and patterning of the hindbrain [19,20,21]. Various *HOX* genes, including *HOXA1*, have also been shown to play a critical role in lung development [22,23,24]. Loss of HOXA1 adversely affects the development of alveoli and pulmonary vessels [22].

Interestingly, HOXA1 and other HOX genes continue to be expressed in adult organs, including the lungs, suggesting their potential contribution to regeneration, repair, and innate immunity [17,25,26,27,28,29]. Genes from the HOXA cluster and HOXA1 have been linked to the self-renewal of hematopoietic stem cells and their differentiation [30,31,32]. Additionally, HOXA1 was also found to promote cell proliferation in lung fibroblasts [32].

Despite the expression of *HOXA1* in adult human lungs [17], and its downregulation in COPD airway basal cells [15], which show defects in airway epithelial generation [10,11], the role of HOXA1 in the generation of the airway epithelium has not been investigated. Given that *HOXA1* is primarily expressed in the basal cells and that basal cells regenerate a fully functional airway epithelium, we sought to investigate the contribution of HOXA1 to airway basal cell proliferation and its effect on the generation of the airway epithelium by using HOXA1 knockdown airway basal cells.

## 2. Materials and Methods

### 2.1. Airway Basal Cells

Basal cells were isolated from bronchial segments of normal donor lungs at the time of lung transplantation or from lungs rejected for transplantation, as described previously [10,15,16]. Collection of the tissue was approved by the Institutional Review Board of Temple University (4407), Philadelphia, PA. Briefly, the bronchial segments were cut into 0.5 to 1 cm pieces and then incubated with trypsin in the presence of antibiotics and antifungal agents for 24 to 48 h at 4 °C. The trypsin was then neutralized with fetal bovine serum, and the epithelial cells were harvested by scraping the mucosal side of the tissue. The released epithelial cells were collected through centrifugation and plated on a collagen-coated cell culture plate using Bronchial Life medium (Lifeline technologies Inc., Frederick, MD, USA) until the cells became 70 to 80% confluent. The cells were collected through trypsinization and stored in a liquid nitrogen tank as P1 cells.

### 2.2. Lentiviral Transduction of Airway Basal Cells

Passage 1 cells were cultured in 6-well collagen-coated dishes until the cells were 50% confluent. The cells were transduced with shERWOOD ultramiR HOXA1 or non-targeting shRNA Lentiviral particles using TransDux Max Lentivirus transduction reagent and Max enhancer reagent (System Biosciences, Palo Alto, CA, USA) according to manufacturer’s instructions using a ratio of cell to lentivirus particle of 0.1. After incubating the cells for 6 h, the transduction medium was replaced with fresh Bronchial Life medium containing 5 mM ROCK inhibitor Y-27632 (Cayman Chemicals, Ann Arbor, MI, USA) and cultured for another 4–6 days or until the cells became 80% confluent. The addition of ROCK inhibitor Y-27632 significantly improved the proliferation of basal cells post-lentiviral transduction. Transient treatment with Y-27632 during the proliferation of airway basal cells does not affect subsequent polarization or differentiation [33]. The medium was changed every other day during this period. The transduced cells were harvested and stored in the liquid nitrogen tank in cryopreservation medium (Bronchial Life medium containing 10% fetal bovine serum and 10% DMSO). To confirm HOXA1 knockdown, the cells were seeded in 6-well plates coated with collagen in Bronchial Life medium containing 5 mM ROCK inhibitor until the cells reached 70 to 80% confluency. They were then treated with 1 µg/mL of doxycycline (Millipore Sigma, St. Louis, MO, USA) and incubated for 72 h to induce HOXA1 shRNA expression. Knockdown of HOXA1 was confirmed through Western blot analysis.

### 2.3. Spheroid Assay

The airway basal cells transduced with *HOXA1* or non-targeting shRNA expressing lentiviral particles were cultured in 6-well collagen coated plates until the cells became 70–80% confluent. Cells were treated with 1 µg/ ml of doxycycline for 48 h to induce knockdown of *HOXA1*. The cells were harvested, prepared in a single cell suspension, and mixed at a 1:1 ratio with 50% Corning^®^ Matrigel^®^ Growth Factor Reduced basement membrane matrix (Millipore Sigma) diluted in Bronchial Life medium without ROCK inhibitor. The final density of cells in the suspension was 10,000 cells/1 mL, and the concentration of Matrigel was 25%. The cell suspension (100 µL) was plated in 6 mm Transwells and cultured for 7 days. The medium in the basolateral chamber was changed every other day. The cultures were imaged using phase contrast microscopy in 2 to 3 random fields per Transwell, and the number and size of the spheroids were quantified using ImageJ software, version 1.54.

### 2.4. Culturing Cells in Transwells

The basal cells transduced with control or HOXA1 shRNA lentiviral vectors were seeded in 6 mm Transwells (50,000 cells/well) and cultured under submerged conditions in Bronchial Life medium containing 5 mM ROCK inhibitor (Y-27632) until the cells became 100% confluent. The cells were then treated with 1 µg/mL of doxycycline and incubated for 72 h to induce HOXA1 knockdown. The cells were imaged under phase contrast microscopy to confirm that the cells were confluent. The cells were then cultured at an air/liquid interface (ALI) in differentiation medium for up to 4 weeks. Some cultures were treated from day 4 to day 10 of culturing at ALI with 2 µM TNF-α receptor antagonist R-7050 (Millipore Sigma) or DMSO.

### 2.5. Quantitation of Apoptotic and Necrotic Cells

The apoptosis and necrotic cells were quantified using an apoptosis assay kit following the manufacturer’s instructions (BD Biosciences, San Jose, CA, USA). Briefly, the normal airway cells transduced with *HOXA1* or non-targeting shRNA containing lentivector were grown in Transwells until the cells reached 100% confluence. The cells were then treated with doxycycline (1 µg/mL) to induce *HOXA1* K/D, and the cells were incubated for 3 days. The cells were harvested along with floating cells and incubated with an antibody to Annexin V, a marker for apoptotic cells and propidium iodide (a marker of necrotic cells and cells with a compromised plasma membrane). The cells were fixed and analyzed using a Symphony flow cytometer. The data were analyzed using FlowJo version 10 (Tree Star, Ashland, OR, USA).

### 2.6. Quantitation of Goblet and Ciliated Cells in Differentiated Cultures

The goblet and ciliated cells in the differentiated cell cultures were detected through immunofluorescence microscopy using antibodies to human tracheobronchial mucins to detect goblet cells and acetylated α-tubulin to detect cilia (Millipore Sigma) using whole cultures, as previously described [11,34]. Briefly, the apical surface of the cell cultures was washed to remove secreted mucin and fixed in cold methanol for −20 °C for 5 min. The cultures were washed with PBS, blocked with 1% BSA and incubated with primary antibodies overnight at 4 °C. The cultures were washed with PBS, and the bound antibodies were detected by using second antibodies conjugated with AlexFlour 488 (for ciliated cells) and AlexFlour 598 (for goblet cells). The cultures were counterstained with DAPI (nuclei), mounted, and imaged through confocal microscopy. The number of goblet cells, ciliated cells, and nuclei (total cells) was counted in 10 random 0.1 mm^2^ areas per culture. The cells that were not positive for either mucin or acetylated α-tubulin antibody were considered to be cell types other than goblet and ciliated cells. The cell types were expressed as the % of total cells.

### 2.7. Transepithelial Resistance and Immunolocalization of E-Cadherin and Occludin Observed Through Confocal Microscopy

Normal airway basal cells transduced with *HOXA1* or non-targeting shRNA (control) expressing lentivector were cultured in collagen-coated Transwells until the cells became approximately 70–80% confluent. Cells were treated with 1 µg/mL of doxycycline for 72 h to induce *HOXA1* knockdown and continued to culture under submerged conditions. After the cells reached 100% confluency, they were maintained at an air/liquid interface in differentiation medium for 10 days. The transepithelial resistance of airway epithelial cell cultures after culturing at ALI for 10 days was determined by using an EVOM VOLT/Ohm meter equipped with EndOhm chambers (World Precision Instruments, Sarasota, FL, USA), as described previously [15,16,35]. Transwells were then fixed in 4% paraformaldehyde, washed with PBS, blocked with 1% BSA, and immuno-stained with antibodies to E-cadherin (BioLegend, San Diego, CA 324012, USA, Cat# 324012) and occludin (Cell Signaling, Danvers, MA, USA, Cat# 91131), as previously described [16]. The bound antibodies were detected by using antirabbit IgG conjugated with Alexa Fluor-594 and antimouse IgG conjugated with Alexa Fluor 488 (Thermofisher Scientific, Waltham, MA, USA). Cell cultures were counterstained with DAPI to detect nuclei and imaged under a confocal microscope. The signal intensities were measured and expressed as pixels per 100 µm^2^.

### 2.8. Immunofluorescence Microscopy for Detection of HOXA1 in the Lungs

Lung tissue at the second and third branching of bronchi was collected from three COPD and three healthy non-smokers’ donor lungs under the approval of Institutional Review Board of Temple University. The tissues were fixed in 10% buffered formalin embedded in paraffin. Five-micron-thick paraffin sections were deparaffinized, incubated for 10 min in boiling 10 mM citrate buffer at pH 6.0, and allowed to cool down to room temperature. The sections were then treated with 3% hydrogen peroxide to quench endogenous peroxidase activity and blocked with 5% normal horse serum containing 0.3% Triton X-100 for 1 h. Slides were washed and incubated with HOXA1 antibody (LifeSpan Biosciences, Shirley, MA, USA, Cat # LS-B13574) for 16 h at 4 °C. The slides were washed, incubated with HRP conjugated secondary antibody, and subjected to the tyramide signal amplification method, as previously described [15,36]. The sections were counterstained with DAPI and imaged using a fluorescence microscope equipped with a CCD camera.

### 2.9. RNA Isolation and qPCR

Total RNA was isolated from TRIZOL lysates of control and HOXA1 knockdown cells after culturing cells for two weeks at ALI using a Direct-zolTM miniprep kit (Zymo research, Irvine, CA, USA). cDNA was synthesized from the total RNA (high-capacity first strand synthesis kit, Thermofisher Scientific) and subjected to real-time qPCR to determine the expression of *Occludin*, *E-cadherin*, *IL-8*, *TNF-α,* and *glyceraldehyde 3-phosphate dehydrogenase* (*GAPDH*) using gene-specific Taqman PCR assays (ThermoFisher Scientific). The expression level of each gene was normalized to the house-keeping gene, *GAPDH*.

### 2.10. ELISA

After culturing the cells transduced with the control or *HOXA1* shRNA containing lentiviral particles for 4 weeks at ALI, Transwells were transferred to a new receiver plate containing fresh medium and incubated for 24 h. The basolateral medium was collected, and IL-6, IL-8, and TNF-α protein levels were measured through ELISA (R&D systems, Minneapolis, MN, USA).

### 2.11. Western Blot Analysis

An equal amount of total protein from the cells was subjected to Western blot analysis with antibodies to 1:1000 diluted HOXA1 (LifeSpan Biosciences), E-cadherin, and occludin. The blots were stripped and re-probed with GAPDH (Millipore Sigma, St. Louis, MO, USA) antibody. The density of the protein bands was determined using ImageJ and expressed as the fold change over GAPDH.

### 2.12. Statistical Analysis

All of the cell culture experiments were performed at least in duplicate and repeated two to three times. Data are presented as the median with the range or the average with the SEM. If the data were normally distributed, statistical significance was assessed using an unpaired *t* Test. If the data were not normally distributed according to Shapiro-Wilk test, the statistical difference between the groups was determined through the Mann–Whitney test. Statistical analysis of the data was conducted using SigmaStat V4 (Grafiti, ST Palo Alto, CA, USA). A *p*-value of ≤0.05 was considered statistically significant.

## 3. Results

### 3.1. Bronchial Epithelium Expresses HOXA1

The paraffin sections of secondary and tertiary bronchi from healthy non-smoker (normal) and COPD patients were immuno-stained with an antibody to HOXA1 to evaluate the localization of the staining. HOXA1 staining was primarily observed in the cells located close to the basement membrane in bronchial sections from normal and COPD subjects (Figure 1A,B). HOXA1 was expressed abundantly in the normal bronchial epithelium, while bronchial sections from patients with COPD showed dramatically reduced expression. The antibody absorbed against the antigen did not show signals, indicating that the antibody specifically recognizes HOXA1 (Figure 1C).

### 3.2. HOXA1 Contributes to Basal Cell Proliferation

HOXA1 is known to regulate cell proliferation in hematopoietic stem cells and in various cancer cell lines. Therefore, we investigated whether HOXA1 also contributes to airway basal cell proliferation. Normal airway basal cells transduced with HOXA1 or scrambled shRNA lentivector were cultured until they reached 70 to 80% confluency. The cells were then treated with doxycycline to induce HOXA1 knockdown (HOXA1 K/D). The knockdown of HOXA1 was confirmed through Western blot analysis from representative cultures (Appendix A). The cells were harvested, suspended into single cells, seeded in Matrigel with reduced growth factors, and cultured for 7 days. Both control and HOXA1 K/D cells cultured in Matrigel for 7 days formed spheroids (Figure 2A,B). There was no difference in the number of spheroids (Figure 2C), indicating an equal number of viable cells seeded in the Matrigel. However, the size of spheroids formed by HOXA1 K/D cells was significantly smaller than that of the control (Figure 2D,E). To estimate the total number of cells, the spheroid cultures were dissociated, and the total number of cells was counted. The number cells were significantly lower in HOXA1 K/D cells than in controls despite seeding equal number of viable cells (Figure 2F). These results indicate that HOXA1 may contribute to airway basal cell proliferation.

### 3.3. Effect of HOXA1 Knockdown on Generation of Airway Epithelium

HOXA1, in addition to contributing to cell proliferation, also participates in the differentiation of hematopoietic cells and preadipocyte differentiation. Therefore, we assessed the contribution of HOXA1 to airway basal cell differentiation into the mucociliary phenotype. To overcome the attenuated cell proliferation in HOXA1 K/D cells, the knockdown of HOXA1 was induced in confluent monolayers of basal cells. Confluent monolayers of control or HOXA1 shRNA lentiviral transduced basal cells growing in Transwells were treated with doxycycline to induce knockdown of HOXA1 in cells. The induction of HOXA1 K/D did not cause cell death, as determined through apoptosis and necrosis assay (Appendix A). The cells also maintained confluency, as determined through phase contrast microscopy (Supplemental Appendix A). To assess the effect of HOXA1 K/D on the generation of the airway epithelium, the control and HOXA1 K/D cells were cultured at ALI for 4 weeks, immuno-stained with antibodies to tracheobronchial mucins and acetylated tubulin, and imaged using a confocal microscope. The number of nuclei (total cells), the ciliated and goblet cells, and the rest of the cells, which were negative for mucin and acetylated tubulin (other cell types), were counted and expressed as the % of total cells per 100 µM^2^ (Figure 3A,B). Compared to the control, HOXA1 K/D cell cultures showed significantly less ciliated cells and more goblet cells and other cell types (Figure 3C).

Abnormally differentiated cell cultures express pro-inflammatory cytokines [34]. Therefore, we examined the levels of IL-6, IL-8, and TNF-α in the basolateral medium of these cultures. The HOXA1 knockdown cultures showed higher levels of all three cytokines compared to the controls (Figure 4A–C).

### 3.4. Role of HOXA1 in Airway Basal Cells’ Polarization

Cell polarization is essential for airway basal cells to differentiate into a mucociliary airway epithelium. Normally, basal cells polarize and develop transepithelial resistance (TER) when confluent monolayers are cultured at ALI for 7 to 10 days [15,16]. Knockdown of *HOXA1* was induced in confluent layers of basal cells transduced with either control or *HOXA1* shRNA lentivector and then cultured at ALI for 10 days. HOXA1 K/D cells showed lower TER than control cells (Figure 5A), indicating a defect in the polarization of cells. The development of TER depends on the formation of adherence and tight junctions. Therefore, we assessed the localization of the adherence junction protein, E-cadherin, and the tight junction protein, occludin. As observed earlier [16], control cells showed intense staining of both E-cadherin and occludin in the intercellular junctions (Figure 5B). In contrast, the intensity of both proteins at intercellular junctions was reduced in HOXA1 K/D cells (Figure 5C). The assessment of the overall signal intensity at the apical surface of the cultures (which represents proteins that present primarily at the intercellular junctions) for E-cadherin and occludin indicated a significant reduction in signal intensities for both proteins in HOXA1 K/D cells compared to the control (Figure 6).

To assess whether the attenuated localization of E-cadherin and occludin is due to reduced protein expression, we assessed the protein levels of E-cadherin and occludin using total proteins through Western blot analysis (Figure 7A–D). The expression of occludin but not E-cadherin was significantly decreased in HOXA1 K/D compared to control cell cultures. To examine whether the reduction of occludin protein occurs at the transcriptional level, we assessed the mRNA levels through qPCR (Appendix A). Again, compared to the control, HOXA1 K/D cell cultures showed lower levels of occludin, but not E-cadherin.

Previously, pro-inflammatory cytokines, including, in particular, TNF-α and IFN-γ, were shown to reduce the transcription of occludin in gut epithelial cells [30]. Therefore, we assessed the protein levels of TNF-α and IFN-γ in the basolateral medium after culturing cells at ALI for 10 days. IFN-γ was below the detection level in both control and HOXA1 K/D cells (data not presented). In contrast, the protein levels of TNF-α were significantly higher in HOXA1 K/D than control cell cultures (Figure 8A).

HOXA1 K/D cells were treated with vehicle or TNF-α antagonist III, R-7050, during the last 6 days of culturing at ALI, and the expression of occludin was assessed through qPCR and Western blot analysis (Figure 8B,C). TNF-α antagonist III treated cells showed significantly higher expression of occludin at both the mRNA and the protein level compared to the vehicle-treated cells, indicating that increased TNF-α in HOXA1 K/D cells may contribute to the observed reduced expression of occludin in these cells.

Taken together, these results indicate that HOXA1 K/D, in addition to cell proliferation, also contributes to the polarization and differentiation of airway basal cells either directly through the inhibition of the pro-inflammatory cytokine, TNF-α, or via the modulation of other molecules.

## 4. Discussion

This study demonstrates for the first time that HOXA1 not only contributes to airway basal cell proliferation but also plays a role in airway epithelial generation by promoting the polarization of cells, an essential prerequisite step for basal cell differentiation into the mucociliary phenotype. Furthermore, HOXA1 regulates the expression of occludin, one of the tight junction proteins.

Previous studies have shown that *HOXA1* mRNA is expressed in adult human but not in adult mouse lungs [17]. In our earlier research, we found that *HOXA1* mRNA is present in airway basal cells, with its expression significantly reduced in COPD patients [15]. For the first time, in this study, we demonstrate that HOXA1 protein is expressed in the bronchial epithelium of healthy non-smokers and dramatically reduced in COPD patients. This expression is primarily localized in cells located close to the basement membrane, which are likely airway basal cells. Considering that HOXA1 plays a role in the self-renewal of hematopoietic stem cells and the differentiation of progenitor cells [30,31], it is plausible that HOXA1, which is primarily expressed in the airway basal cells of the normal bronchial epithelium, contributes to epithelial repair and regeneration. Consistent with this notion, HOXA1 K/D in the confluent monolayer of basal cells leads to the generation of an abnormal airway epithelium with goblet cell metaplasia and a pro-inflammatory phenotype. COPD airway basal cells, which show downregulation of *HOXA1* expression, generate an airway epithelium with a similar abnormal phenotype [10,11].

Both cell proliferation and polarization are essential for airway epithelial regeneration/repair from airway basal cells. To examine the effect of HOXA1 in cell proliferation and polarization, we developed an inducible *HOXA1* knockdown system, which is essential for expanding the cells following lentiviral transduction and also to induce HOXA1 K/D at different growth stages. Inducing HOXA1 K/D in proliferating cells resulted in smaller spheroids, although there was no change in the total number of spheroids. Because each spheroid originates from a single cell, the size of the sphere directly correlates with cell proliferation, indicating that HOXA1 may contribute to cell proliferation/the self-renewal of airway basal cells.

HOXA1 may regulate cell proliferation through multiple mechanisms in normal cells. For instance, in human granulosa cells, downregulation of HOXA1 decreased cell proliferation by increasing apoptosis and inducing mitochondrial dysfunction [31]. HOXA1 knockdown in fibroblasts or HEK293 cells resulted in cell cycle arrest at the G0/G1 phase, suggesting that HOXA1 may influence cell proliferation by regulating cell cycle progression [32]. However, the specific mechanisms of HOXA1 in the self-renewal and proliferation of airway basal cells remain to be uncovered. Our ongoing studies suggest that HOXA1 may regulate cell proliferation via the prevention of premature contact inhibition of locomotion.

Surprisingly, our findings also suggested a role for HOXA1 in airway epithelial cell polarization. This was not due to a defect in cell proliferation in HOXA1 K/D cells, as knockdown was induced after the cells reached 100% confluency, when cells cease to proliferate and start to polarize. Moreover, no cell death or change in confluency was observed, implying that the observed attenuated TER is due to the reduced expression of HOXA1 in HOXA1 K/D cells. The decrease in TER was accompanied by the reduced localization of E-cadherin and occludin to the tight junctions. Intriguingly, we found that HOXA1 regulated the expression of occludin but not E-cadherin at the transcriptional and translational levels. Neither occludin nor E-cadherin has a HOXA1 binding site in their promoter region, indicating that HOXA1 may regulate occludin expression indirectly.

It may seem counterintuitive for HOXA1, which is vital for basal cell proliferation or self-renewal, to also partake in cell polarization when cell proliferation is minimal. However, HOXA1 may play a bimodal role; that is, once the cells reach confluence, it may downregulate the signaling pathways involved in cell proliferation and upregulate the pathways involved in polarization. Previously, HOXA1 has been demonstrated to interact with TNF receptor associated factors (TRAF1 and TRAF2), which play a role in TNF-α signaling [37,38]. TNF-α and IFN-γ negatively regulate occludin and other tight junction proteins, but not E-cadherin at the transcriptional level [39]. Intriguingly, we observed that HOXA1 K/D causes increased expression of TNF-α, and inhibition of TNF-α enhances the expression of occludin mRNA and protein. These observations indicate that HOXA1 may be required for blocking the expression of TNF-α and for inhibiting TNF-α-induced downregulation of occludin expression.

Cell cultures generated from HOXA1 K/D airway basal cells also showed abnormal differentiation with goblet cell hyperplasia. This is not surprising, as polarization is a prerequisite for differentiation. Previously, we demonstrated that airway basal cells that show impaired polarization or attenuation in polarization differentiate abnormally, showing more goblet cells and less ciliated cells [34]. In another study, polarization was shown to direct epithelial cell fate towards ciliated cells via inhibition of the Yap pathway [40]. The polarization of cells has also been shown to be required for genes that regulate multiciliated cell differentiation [41]. Therefore, the reduced number of ciliated cells in HOXA1 may be due to a loss of molecular cues that promote ciliated cell differentiation.

In conclusion, our findings indicate that HOXA1 is not only important for airway basal cell proliferation but also for polarization and thus the differentiation of these cells. Downregulation of HOXA1 in basal cells can therefore influence the overall structure of the airway epithelium with goblet cell metaplasia. Based on these observations, we speculate that the abnormal phenotype of airway epithelial cell cultures generated from COPD airway basal cells may be due to the downregulation of *HOXA1*, which will be investigated in future studies.

## Figures and Tables

**Figure 1 cells-14-00549-f001:**
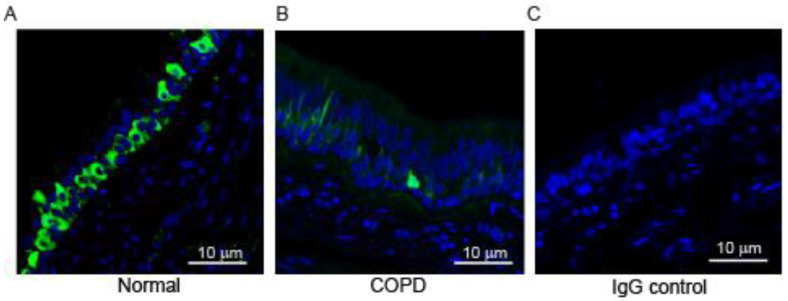
HOXA1 is expressed in the bronchial epithelium, and expression is reduced in patients with COPD. Green, HOXA1; blue, nuclei. (**A**,**B**) Lung sections from normal and COPD subjects stained with HOXA1 antibody. (**C**) Normal lung section stained with isotype control IgG. The images are representative of lung sections from 3 healthy non-smokers.

**Figure 2 cells-14-00549-f002:**
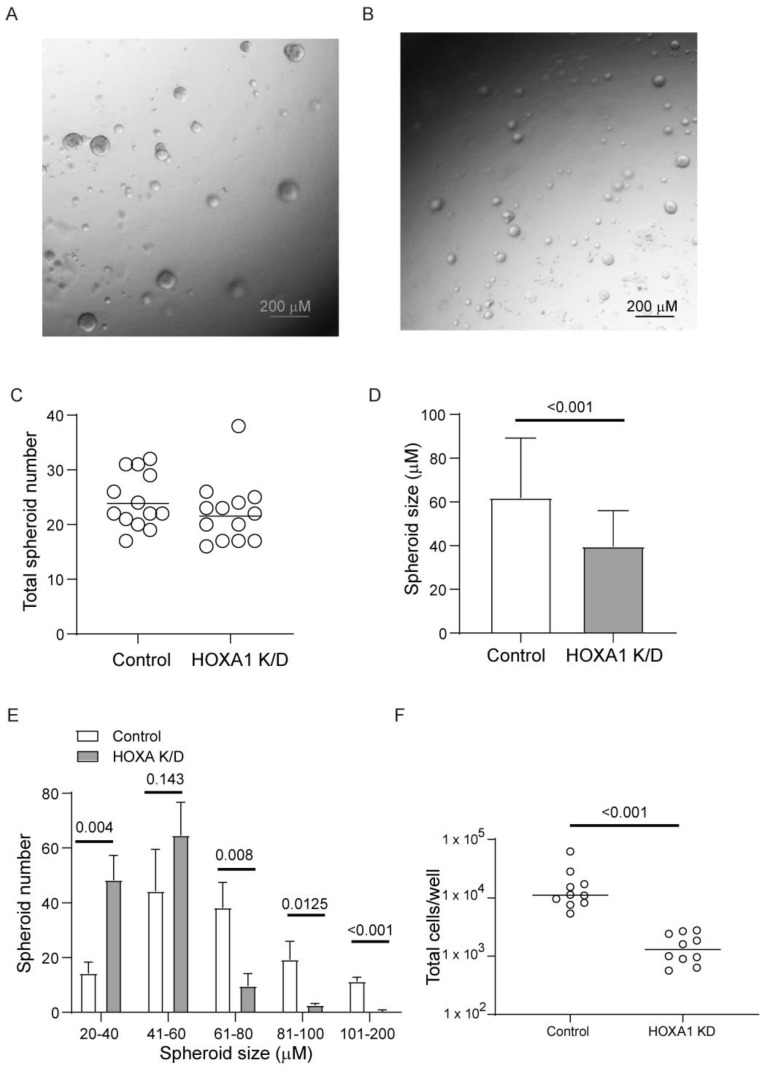
HOXA1 K/D airway basal cells show attenuated proliferation compared to control cells. (**A**,**B**) Representative images of control and HOXA1 K/D cells, respectively. (**C**) Data represent the range with the median from two independent experiments performed in triplicate. (**D**,**E**) Data represent the size of the spheres averaged from two independent experiments performed in triplicate (*t* test). Two to three random fields per culture were used to determine the number and size of the spheres. (**F**) Spheres were dissociated from each well, and the total number of cells was estimated. Data represent the range with the median from two independent experiments performed in triplicate (Mann–Whitney test).

**Figure 3 cells-14-00549-f003:**
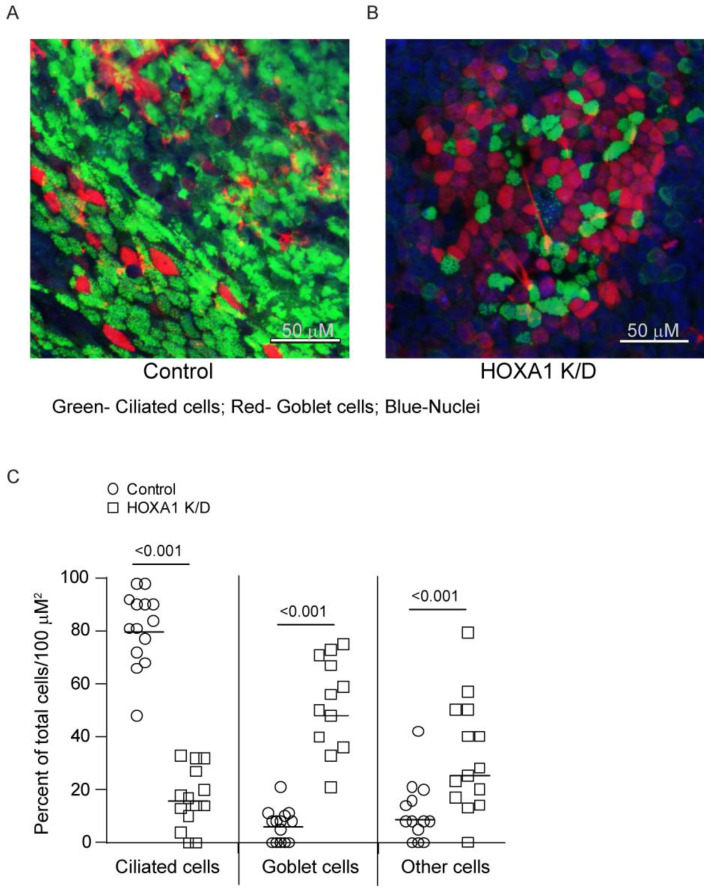
Mucociliary-differentiated airway epithelial cell cultures generated from HOXA1 K/D basal cells show more goblet cells and less ciliated cells than the control. (**A**,**B**) Confocal immunofluorescence images showing ciliated cells (green), goblet cells (red), and nuclei (blue) in control and HOXA1 cultures, respectively. Images are representative of 3 independent experiments. (**C**) The number of goblet cells, ciliated cells (green), and other cell types (DAPI) was counted per 100 µm^2^, and the data are presented as the range with the median.

**Figure 4 cells-14-00549-f004:**
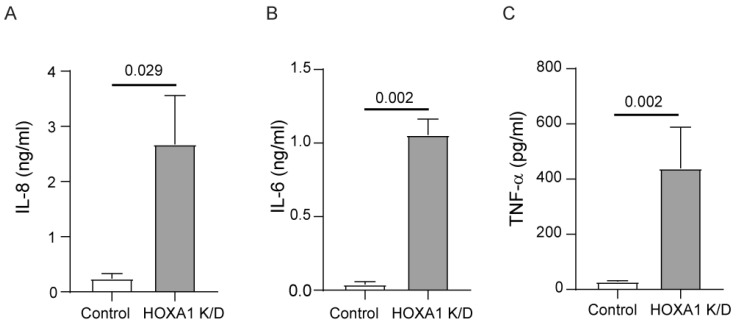
HOXA1 K/D cell cultures show higher levels of pro-inflammatory cytokines than control cell cultures. Cytokine levels were measured in the basolateral medium from the mucociliary-differentiated cultures through ELISA. (**A**) IL-8, (**B**) IL-6, and (**C**) TNF-α. Results represent the mean ± S.D. calculated from 3 independent experiments (*n* = 3; unpaired *t* test).

**Figure 5 cells-14-00549-f005:**
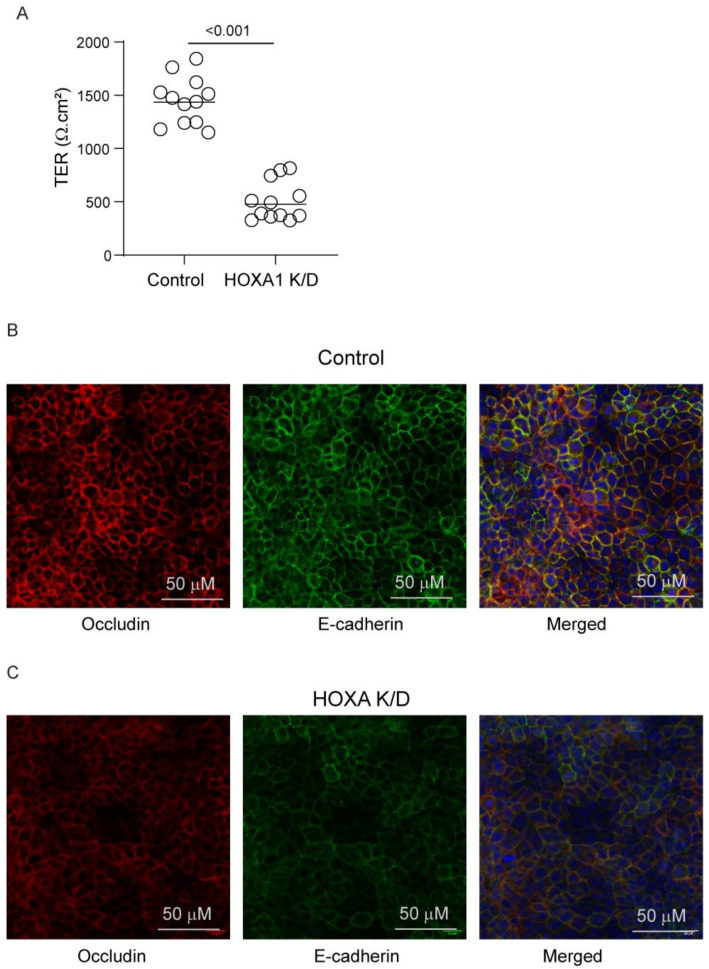
HOXA1 K/D airway basal cells show a defect in tight junction formation. (**A**) The TER was measured, and the data represent the range with the median from 3 independent experiments performed in quadruplicate (*n* = 12, Mann–Whitney test, *p* ≤ 0.05, different from control). (**B**,**C**) Cells were fixed, blocked with BSA, and incubated with antibodies to occludin and E-cadherin. The bound antibodies were detected using antirabbit IgG conjugated with Alexa Fluor-594 (occludin) and antimouse IgG conjugated with Alexa Fluor 488 (E-cadherin). Cell cultures were counterstained with DAPI (blue) to detect nuclei and imaged under a confocal microscope. Images are representative of 3 independent experiments.

**Figure 6 cells-14-00549-f006:**
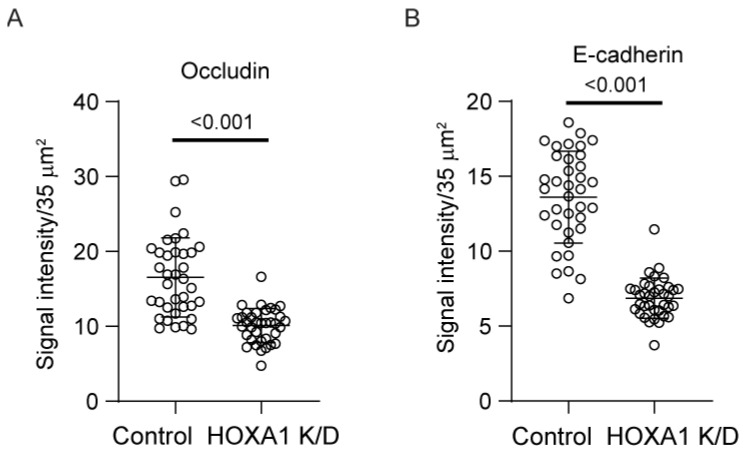
Quantitation of signal intensity of E-cadherin and occludin in polarized airway epithelial cells. Polarized cell cultures immuno-stained with occludin and E-cadherin were imaged under confocal microscopy, and signal intensities were measured as pixels in 10–12 random fields per Transwell; data are presented as the median with the range (*n* = 3, Mann–Whitney test).

**Figure 7 cells-14-00549-f007:**
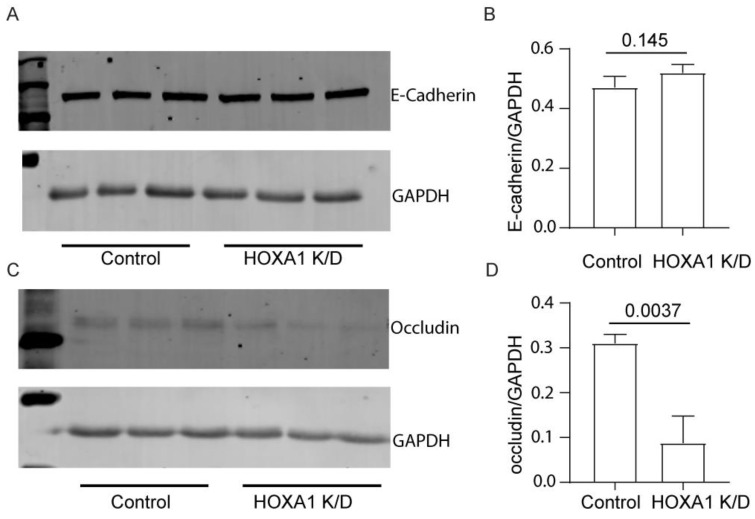
Expression of occludin is decreased in HOXA1 K/O cells at the protein level. (**A**,**C**) Western blot showing the expression of E-cadherin and occludin, respectively. (**B**,**D**) The band densities were quantified using ImageJ and expressed as a fold change over GAPDH. Data in B and D are presented as the mean ± S.D. calculated from 3 independent experiments (*n* = 3, unpaired *t* test).

**Figure 8 cells-14-00549-f008:**
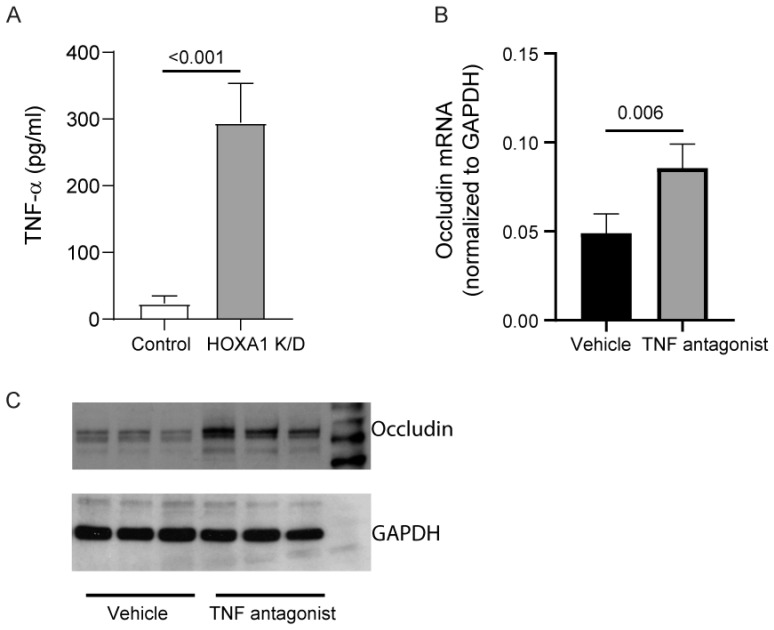
Inhibition of TNF-α improves occludin expression in HOXA1 K/D cell cultures. (**A**) The TNF-α level was measured in the basolateral medium of the control and HOXA1 K/D cells cultured at ALI for two weeks. (**B**) TNF-α mRNA levels were measured in HOXA1 K/D cell cultures treated with 2 µM TNF-α antagonist III or vehicle, and the results represent TNF-α expression levels normalized to GAPDH. (**C**) Western blot analysis of the total protein isolated from HOXA1 K/D cell cultures treated with 2 µM TNF-α antagonist III or vehicle. Data in A and B are presented as the mean ± S.D. calculated from 3 independent experiments (*n* = 3, unpaired *t* test).

## Data Availability

All data are presented in the manuscript.

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
