# Peer review of "Role of Homeobox A1 in Airway Epithelial Generation from Human Airway Basal Cells"

_cells, 2025, doi:10.3390/cells14070549_

Round 1

Reviewer 1 Report

Comments and Suggestions for Authors

Tabashi et al. describe the role of the HOXA1 transcription factor in airway epithelial cell proliferation and differentiation. Using the shRNA knockdown model, they reveal that HOXA1 loss causes decreased airway cell proliferation, ciliated cells, and goblet cell hyperplasia. While the manuscript title and conclusion suggest HOXA1 modulates regeneration (the recovery process after injury), there is no direct data from this current to support this inference. I have provided line-by-line feedback below that needs revision by the authors: 

Line 2: The role of homeobox A1 in basal cells on human airway epithelial proliferation and differentiation. Revise the title according to the data from the study.

Line 66: Is it that the role of HOXA1 has not been defined or that the regulation of the epithelial repair process by the transcription factor remains unknown? Please revise the sentence for clarity.

Lines 73-74: Briefly describe the benchtop isolation procedure here.

Line 155: "After culturing the cells..." State the experimental groups for the ELISA assay.

Line 161: delete 'g' in occluding

Move sentences 2–4 in Fig. 1 to the IF subsection in the method section. Revise accordingly for the subsequent legends where necessary.

Line 227: Did you induce injury to assess the HOXA1 effect during regeneration? Fig. 2A showed no difference in cell number, and HOXA1 K/D did not cause cell death. If so, revise your result here to indicate HOXA1 impairs differentiation of ciliated cells. I assume you want to write knockdown instead of K/O?

Line 235: update your reference with proper citation style

Line 341: "Previously, occludin expression has been shown to reduce transcription of occludin in
gut epithelial cells, in particular TNF-α and IFN-γ [30]." This sentence is confusing. Do you want to type cytokine expressions or what? Please revise this sentence.

Line 358: "... other molecules." Specify which molecule is revealed in this study, i.e., via the TNF-α molecule.

Line 361: "... epithelial regeneration..." I would suggest you focus on the data you generated in this study. Here, the results show impaired cilia epithelial cell differentiation contributing to delayed barrier integrity as shown with TEER.

Line 375: "...regeneration..."? generation might be appropriate in the context of your current study. 

Line 384: "Since each spheroid originates from a single cell," Please clarify how you generated the spheroid in the spheroid assay in the method section. Is the spheroid generated from dissociated single basal stem cells or from monolayer fragments for clarity?

Comments on the Quality of English Language

 Except for a few sentences that need clarity, I was able to read the article without stress. 

Author Response

Tabashi et al. describe the role of the HOXA1 transcription factor in airway epithelial cell proliferation and differentiation. Using the shRNA knockdown model, they reveal that HOXA1 loss causes decreased airway cell proliferation, ciliated cells, and goblet cell hyperplasia. While the manuscript title and conclusion suggest HOXA1 modulates regeneration (the recovery process after injury), there is no direct data from this current to support this inference. I have provided line-by-line feedback below that needs revision by the authors: 

We thank the reviewer for the constructive criticisms, which helped us to improve the manscuript.

  1. 1. Line 2: The role of homeobox A1 in basal cells on human airway epithelial proliferation and differentiation. Revise the title according to the data from the study.

We agree with the reviewer.  The data presented in the manuscript the role of HOXA1 in airway epithelial generation rather than regeneration.  Therefore, we have modified the title accordingly. 

  1. 2. Line 66: Is it that the role of HOXA1 has not been defined or that the regulation of the epithelial repair process by the transcription factor remains unknown? Please revise the sentence for clarity.

In this manuscript, we provide data showing the overall role of HOXA1 rather than transcriptional regulation of repair process of airway epithelium.   Therefore, we feel that “HOXA1 role” suits better than transcriptional regulation and the retained the sentence. 

  1. 3. Lines 73-74: Briefly describe the benchtop isolation procedure here.

We have now provided brief description on isolation of basal cells from the airways (Ln 103-108)

  1. 4. Line 155: "After culturing the cells..." State the experimental groups for the ELISA assay.

We have added cells transduced with control or HOXA1 shRNA ………….” (Ln. 221)

  1. Line 161: delete 'g' in occluding.

Thank you for catching the spelling mistake.  We have removed the “g”.

  1. 6. Move sentences 2–4 in Fig. 1 to the IF subsection in the method section. Revise accordingly for the subsequent legends where necessary.

We have removed the methods from all the figure legends as suggested. Ln. 182-187; 202-209;

  1. Line 227: Did you induce injury to assess the HOXA1 effect during regeneration? Fig. 2A showed no difference in cell number, and HOXA1 K/D did not cause cell death. If so, revise your result here to indicate HOXA1 impairs differentiation of ciliated cells. I assume you want to write knockdown instead of K/O?

No, we did not induce injury, therefore we have replaced “regeneration” with generation.  We would like to clarify that Figure 2A shows number of spheroids.  Each live cell will generate one spheroid and the size of the spheroid is an indication of proliferation and not the number.  Since we seeded similar number of viable cells, both control and HOXA1 K/D cells show similar number of spheroids (Figure 2A).  However, the size of the spheroids is much smaller in HOXA1 K/D cells than in control indicating attenuated proliferation in HOXA1 K/D cells.  Figure 2F shows the total number of cells isolated from the spheroid cultures, which show significantly less total cells in HOXA1 K/D than control confirming that cell proliferation is attenuated in HOXA1 K/D.    

Corrected K/O to K/D.

  1. 8. Line 235: update your reference with proper citation style

Thank you, changed it journal style.

  1. Line 341: "Previously, occludin expression has been shown to reduce transcription of occludin in
    gut epithelial cells, in particular TNF-α and IFN-γ [30]." This sentence is confusing. Do you want to type cytokine expressions or what? Please revise this sentence.

Sorry for the confusion.  It is not occludin, but cytokine.  We revised the sentence.

  1. 11. Line 358: "... other molecules." Specify which molecule is revealed in this study, i.e., via the TNF-α molecule.

We have revised the sentence adding TNF-α that was revealed in our studies. Ln.439

  1. Line 361: "... epithelial regeneration..." I would suggest you focus on the data you generated in this study. Here, the results show impaired cilia epithelial cell differentiation contributing to delayed barrier integrity as shown with TEER.

The data show that HOXA1 not only is required for proliferation of cells, but also for generation of epithelium.  The defect in polarization may contribute to generation of abnormal airway epithelium with increased number of goblet cells.  We have replaced “regeneration” with generation through out the manuscript.

  1. 13. Line 375: "...regeneration..."? generation might be appropriate in the context of your current study. 

Agreed.  Please see the response for comment 12.

  1. 14. Line 384: "Since each spheroid originates from a single cell," Please clarify how you generated the spheroid in the spheroid assay in the method section. Is the spheroid generated from dissociated single basal stem cells or from monolayer fragments for clarity?

We generated spheroids from dissociated single cell suspension of airway basal cells. Ln. 135, 264, 272

Comments on the Quality of English Language

 Except for a few sentences that need clarity, I was able to read the article without stress. 

Thank you.  We have revised the manuscript and also the manuscript was edited by professional editing services.

Reviewer 2 Report

Comments and Suggestions for Authors
  1. The introductory sentence in the Abstract need editing. Give an introduction relevance to the disease significance of the study.
  2. Please provide your hypothesis and aim of the study clearly in the abstract and Introduction. The rationale of the study is not clearly stated. Make a separate paragraph for that. Clearly state what is known and what novelty is in the current study. Also please mention why HOXA1 is targeted and why it matters for the disease relevance.
  3. What is the use of ROCK inhibitor in the transduction method?
  4. Mention clearly why basal airway cells are important in lung disease and why this cell type is being focused?
  5. The background written in the Introduction section is not well organized and a bit lengthy. Please rewrite it and make it divided in paragraphs highlighting background/what is known in literature, what is the knowledge gap questioned that is being focused for this study, what is the strategy and aim to address the gap and what is the future prospect.
  6. What is the purpose of goblet and ciliated cells in differentiated culture?
  7. What is the status of HOXA1 expression in COPD patients’ bronchial epithelial? In figure 1 what is the source of the lung tissue section? Please mention is it from healthy control and patients?
  8. In fig 1, How the author did measured proliferation of basal cell? Please use BrdU proliferation assay or use proliferation marker to proof this hypothesis?
  9. How long the airway basal cells are alive in the in vitro culture?
  10. The authors have mentioned, “Induction of HOXA1 did not cause cell death and the cells maintained the confluency (data not shown).” I believe this is very important data which needs to be shown in this manuscript to understand how HOXA1 is affecting cell death/apoptosis. Please use tunel assay to show this.
  11. The title of the each subsection of results need to be changed. Highlight the outcome instead in the title.
  12. What is the meaning of green and red maker/antibody used in Fig 3 to detect the cell types and why these cell types are being focused? Its very unclear.
  13. In Fig 6 how the quantification has been done? Why there are so many data points?
  14. The novelty of the study has to be focused very clearly in the Discussion along with the future direction in the clinical field if any.
Comments on the Quality of English Language

The manuscript is not well written. It needs improvement.

Author Response

  1. The introductory sentence in the Abstract need editing. Give an introduction relevance to the disease significance of the study.

We have edited the abstract as suggested by the reviewer adding disease specific relevance. Ln. 12-13

  1. Please provide your hypothesis and aim of the study clearly in the abstract and Introduction. The rationale of the study is not clearly stated. Make a separate paragraph for that. Clearly state what is known and what novelty is in the current study. Also please mention why HOXA1 is targeted and why it matters for the disease relevance.

Both abstract and introduction has been revised to state the background and the rationale for conducting the study.  We have also added description on the disease relevance. Ln. 63-64-45, 70-71, 91-94.

  1. What is the use of ROCK inhibitor in the transduction method?

The Rock inhibitor inhibitor Y27632 is routinely added to maintain enhance the proliferation of airway basal cells.  It is particularly important in the lentivirus transduced cells, since it improves the posttransduction cell proliferation efficiency, which is necessary for expanding the cells for subsequent experiments. Therefore, we added Y-27632 after transducing the cells with lentivirus. We have added this information in the manuscript. Ln. 119-122.

  1. Mention clearly why basal airway cells are important in lung disease and why this cell type is being focused?

Thank you for your suggestion.  We added this information in paragraphs 1 and 2. Ln. 35-45, 50-54.

  1. The background written in the Introduction section is not well organized and a bit lengthy. Please rewrite it and make it divided in paragraphs highlighting background/what is known in literature, what is the knowledge gap questioned that is being focused for this study, what is the strategy and aim to address the gap and what is the future prospect.

The introduction has been revised substantially quoting appropriate references and the gap in the literature and the rationale for the current study. (marked in red fonts)

  1. What is the purpose of goblet and ciliated cells in differentiated culture?

The goblet and ciliated cells play a crucial role in the mucociliary clearance of inhaled pathogens and particulates.  This is described in the first paragraph of the introduction. Ln 37-41.

  1. What is the status of HOXA1 expression in COPD patients’ bronchial epithelial? In figure 1 what is the source of the lung tissue section? Please mention is it from healthy control and patients?

In COPD bronchial epithelia, expression of HOXA1 is dramatically reduced and we have now added this to the manuscript.  The figure provided was from a healthy subject. Figure 1B, Ln 247-248.

  1. In fig 1, How the author did measured proliferation of basal cell? Please use BrdU proliferation assay or use proliferation marker to proof this hypothesis?

We are sorry for the confusion.  Figure 1 shows the expression of HOXA1 in the fixed lung section from a normal subject.  We show that HOXA1 is primarily expressed in the cells closer to the basement membrane of epithelium and likely airway basal cells.  Since the airway basal cells are quiescent under homeostasis is it unlikely to see proliferating cells even if we use ki67 a marker for actively proliferating cells.

  1. How long the airway basal cells are alive in the in vitro culture?

The airway basal cells can be maintained in culture for up to 3 to 5 passages.  However, after passage 3, the basal cells show slower growth, change in morphology.   Therefore we use the cells in passage 1 or 2.

  1. The authors have mentioned, “Induction of HOXA1 did not cause cell death and the cells maintained the confluency (data not shown).” I believe this is very important data which needs to be shown in this manuscript to understand how HOXA1 is affecting cell death/apoptosis. Please use tunel assay to show this.

I think the reviewer is talking about “induction of HOXA1 shRNA”.  We had measured the cell death using apoptosis kit and we now present those results as supplemental figure (Figure S2).  We had also imaged the cells by phase contrast microscopy to make sure the cells are confluent.  This data is now included in new Supplemental Figure (Figure S2C) 

  1. The title of the each subsection of results need to be changed. Highlight the outcome instead in the title.

We have revised the subtitles as suggested by the reviewer

  1. What is the meaning of green and red maker/antibody used in Fig 3 to detect the cell types and why these cell types are being focused? Its very unclear.

The red color represent goblet cells and green ciliated cells in Figure 3.  We used antibodies to acetylated-α tubulin to detect cilia and mucin antibody to detect goblet cells.  These cells play extremely role in the mucociliary function, therefore we focused on these cells. We have added more information on these cell types in the first paragraph of the introduction.

  1. In Fig 6 how the quantification has been done? Why there are so many data points?

In this figure we measured the intensity of occludin and E-cadherin in 10 to 12 random fields in 3 different cultures.  Since the expression of these proteins varied widely within the culture we presented all the data points.

  1. The novelty of the study has to be focused very clearly in the Discussion along with the future direction in the clinical field if any.

Thank you for the suggestion.  We have modified the discussion to focus on the current finding and future direction. Ln. 501-509, 513-516.

Comments on the Quality of English Language

The manuscript is not well written. It needs improvement.

Please see the response to last question from Reviewer 1.